# A Heavy-Tailed Algebra for Probabilistic Programming

**Feynman Liang**
Department of Statistics
University of California, Berkeley
feynman@berkeley.edu

**Liam Hodgkinson**
School of Mathematics and Statistics
University of Melbourne, Australia
lhodgkinson@unimelb.edu.au

**Michael W. Mahoney**
ICSI, LBNL, and Department of Statistics
University of California, Berkeley
mmahoney@stat.berkeley.edu

## Abstract

Despite the successes of probabilistic models based on passing noise through neural networks, recent work has identified that such methods often fail to capture tail behavior accurately—unless the tails of the base distribution are appropriately calibrated. To overcome this deficiency, we propose a systematic approach for analyzing the tails of random variables, and we illustrate how this approach can be used during the static analysis (before drawing samples) pass of a probabilistic programming language (PPL) compiler. To characterize how the tails change under various operations, we develop an algebra which acts on a three-parameter family of tail asymptotics and which is based on the generalized Gamma distribution. Our algebraic operations are closed under addition and multiplication; they are capable of distinguishing sub-Gaussians with differing scales; and they handle ratios sufficiently well to reproduce the tails of most important statistical distributions directly from their definitions. Our empirical results confirm that inference algorithms that leverage our heavy-tailed algebra attain superior performance across a number of density modeling and variational inference (VI) tasks.

## 1   Introduction

Within the context of modern probabilistic programming languages (PPLs), recent developments in functional programming [51], programming languages [3], and deep variational inference (VI) [4] combine to facilitate efficient probabilistic modelling and inference. But despite the broadening appeal of probabilistic programming, common pitfalls such as mismatched distribution supports [32] and non-integrable expectations [53, 55, 60] remain uncomfortably commonplace and remarkably challenging to address. In particular, heavy-tailed distributions arise in a wide range of statistical applications and are known to present substantial technical challenges [37, 55, 60]. Recent innovations aiming to improve PPLs have automated verification of distribution constraints [32], tamed noisy gradient estimates [16] as well as unruly density ratios [53, 55], and approximated high-dimensional distributions with non-trivial bulks [39]. To address the issue of heavy-tailed targets, approaches which initialize with non-Gaussian tails have been proposed [25, 33]. However, these methods typically require the use of optimization and/or sampling strategies to estimate the tails of the target distribution. Such strategies are often unstable, or fail to allow for a sufficiently wide array of possible tail behaviours.

37th Conference on Neural Information Processing Systems (NeurIPS 2023).

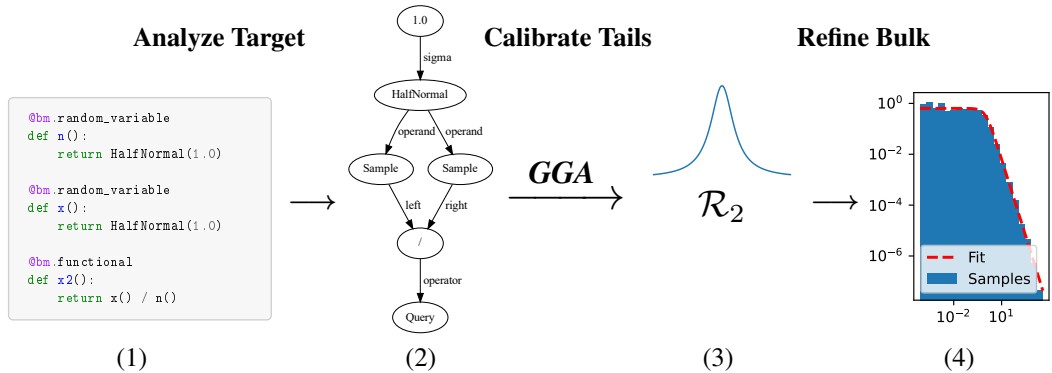

Figure 1: Our heavy-tailed algebra ensures that the tails of density estimators and variational approximations are calibrated to those of the target distribution. Here, a generative model expressed in a PPL (1) is analyzed using the GGA *without drawing any samples* (2) to compute the tail parameters of the target. A representative distribution with calibrated tails is chosen for the initial approximation (3), and a learnable tail-invariant Lipschitz pushforward (see bottom of Table 1, and Theorem 2) is optimized (4) to correct the bulk approximation.

Motivated by this, we introduce the first procedure for static analysis of a probabilistic program that automates analysis of target distributions' tails. In addition, we show how tail metadata obtained from this procedure can be leveraged by PPL compilers to generate inference algorithms which mitigate a number of pathologies. For example, importance sampling estimators can exhibit infinite variance if the tail of the approximating density is lighter than the target; most prominent black-box VI methods are incapable of changing their tail behaviour from an initial proposal distribution [25, 33]; and Monte-Carlo Markov Chain (MCMC) algorithms may also lose ergodicity when the tail of the target density falls outside of a particular family [42]. All of these issues could be avoided if the tail of the target is known before runtime.

To classify tail asymptotics, we propose a three-parameter family of distributions which is closed under most typical operations. This family is based on the generalized Gamma distribution (Equation (2)), and it interpolates between established asymptotics on sub-Gaussian random variables [31] and regularly varying random variables [35]. Algebraic operations on random variables can then be lifted to computations on the tail parameters. This results in a *heavy-tailed algebra* that we designate as the *generalized Gamma algebra (GGA)*. Through analyzing operations like $X + Y$, $X^2$, and $X/Y$ at the level of densities (e.g., additive convolution $p_X \oplus p_Y$), the tail parameters of a target density can be estimated from the parameters of any input distributions using Table 1.

Operationalizing our GGA, we propose a tail inferential static analysis strategy analogous to traditional type inference. GGA tail metadata can be used to diagnose and address tail-related problems in downstream tasks, such as employing Riemannian-manifold methods [17] to sample heavy tails or preemptively detect unbounded expectations. Here, we consider density estimation and VI where we use the GGA-computed tail of the target density to calibrate our density approximation. When composed with a learnable Lipschitz pushforward map (Section 3.2), the resulting combination is a flexible density approximator with tails provably calibrated to match those of the target.

**Contributions.** Here are our main contributions.

- We propose the generalized Gamma algebra (GGA) as an example of a heavy-tailed algebra for probability distributions. This extends prior work on classifying tail asymptotics, while including both sub-Gaussian / sub-exponentials [31] as well as power-law / Pareto-based tail indices [11]. Composing operations outlined in Table 1, one can compute the GGA tail class for downstream random variables of interest.

- We implement GGA as abstract interpretation during the static analysis phase of a PPL compiler. This unlocks the ability to leverage GGA metadata in order to better tailor MCMC and VI algorithms produced by a PPL.

- Finally, we demonstrate that density estimators which combine our GGA tails with neural networks (autoregressive normalizing flows [39] and neural spline flows [15]) simultaneously achieves calibrated tails without sacrificing good bulk approximation.

## 2 The Generalized Gamma Algebra

First, we formulate our heavy-tailed algebra of random variables that is closed under most standard elementary operations (addition, multiplication, powers). The central class of random variables under consideration are those with tails of the form in Definition 1.

**Definition 1.** A random variable $X$ is said to have a *generalized Gamma tail* if the Lebesgue density of $|X|$ satisfies

$$p_{|X|}(x) = x^\nu e^{-\sigma x^\rho}, \qquad \text{as } x \to \infty, \tag{1}$$

for some $c > 0$, $\nu \in \mathbb{R}$, $\sigma > 0$ and $\rho \in \mathbb{R}$. Denote the set of all such random variables by $\mathcal{G}$.

Consider the following equivalence relation on $\mathcal{G}$: $X \equiv Y$ if and only if $0 < p_{|X|}(x)/p_{|Y|}(x) < +\infty$ for all sufficiently large $x$. The resulting equivalence classes can be represented by their corresponding parameters $\nu, \sigma, \rho$. Hence, we denote the class of random variables $X$ satisfying Equation (1) by $(\nu, \sigma, \rho)$. In the special case where $\rho = 0$, for a fixed $\nu < -1$, each class $(\nu, \sigma, 0)$ for $\sigma > 0$ is equivalent, and is denoted by $\mathcal{R}_{|\nu|}$, representing *regularly varying* tails. Our algebra operates on these equivalence classes of $\mathcal{G}$, characterizing the change in tail behaviour under various operations. To incorporate tails which lie outside of $\mathcal{G}$, we let $\mathcal{R}_1$ incorporate *super-heavy tails*, which denote random variables with tails heavier than any random variable in $\mathcal{G}$. All operations remain consistent with this notation. Likewise, we let $\mathcal{L}$ denote *super-light tails*, which are treated in our algebra as a class where $\rho = +\infty$ (effectively constants).

Equation (1) and the name of the algebra are derived from the generalized Gamma distribution.

**Definition 2.** Let $\nu \in \mathbb{R}$, $\sigma > 0$, and $\rho \in \mathbb{R}\backslash\{0\}$ be such that $(\nu + 1)/\rho > 0$. A non-negative random variable $X$ is *generalized Gamma distributed* with parameters $\nu, \sigma, \rho$ if it has Lebesgue density

$$p_{\nu, \sigma, \rho}(x) = c_{\nu, \sigma, \rho} x^\nu e^{-\sigma x^\rho}, \qquad x > 0, \tag{2}$$

where $c_{\nu, \sigma, \rho} = \rho \sigma^{(\nu+1)/\rho}/\Gamma((\nu + 1)/\rho)$ is the normalizing constant.

The importance of the generalized Gamma form arises due to a combination of two factors:

  (i) The majority of interesting continuous univariate distributions with infinite support satisfy Equation (1), including Gaussians ($\nu = 0$, $\rho = 2$), gamma/exponential/chi-squared ($\nu > -1$, $\rho = 1$), Weibull/Frechet ($\rho = \nu + 1$), and Student $T$/Cauchy/Pareto ($\mathcal{R}_\nu$). A notable exception is the log-normal distribution (see Example 8 in Appendix C).

  (ii) The set $\mathcal{G}$ is known to be closed under additive convolution, positive powers, and Lipschitz functions. We prove it is closed under multiplicative convolution as well. This covers the majority of elementary operations on independent random variables. Reciprocals, exponentials and logarithms comprise the only exceptions, however, we will introduce a few "tricks" to handle these cases as well.

The full list of operations in GGA is compiled in Table 1 and described in detail in Appendix A. GGA classes for common probability distributions are provided in Appendix B. All operations in the GGA can be proven to exhibit identical behaviour with their corresponding operations on random variables, with the sole exception of reciprocals (marked by †), where additional assumptions are required. The asymptotics for operations marked with an asterisk are novel to this work. For further details, refer to Appendix A.

**Repeated applications.** Provided independence holds, composition of operations in the GGA remain consistent unless one applies Lipschitz functions, logarithms, or exponentials. If one of these operations is applied, the tail becomes an upper bound, which remains consistent under addition, multiplication, and powers, but not reciprocals. Given that we are working with a fixed class of tails, such behavior is inevitable, and it is possible to perform a sequence of operations for which the tail no longer becomes accurate.

**Posterior distributions.** A primary application of PPLs is to perform Bayesian inference. To cover this use case, it is necessary to prescribe a procedure to deal with *posterior distributions*. Consider a setup where a collection of random variables $X_1, \ldots, X_n$ are dependent on corresponding latent random elements $Z_1, \ldots, Z_n$ as well as a parameter $\theta$ through functions $f_i$ by $X_i = f_i(\theta, Z_i)$. For

| | | |
|---|---|---|
| **Ordering** | $(\nu_1, \sigma_1, \rho_1) \le$ $(\nu_2, \sigma_2, \rho_2)$ | $\iff \limsup_{x \to \infty} \frac{x^{\nu_1} e^{-\sigma_1 x^{\rho_1}}}{x^{\nu_2} e^{-\sigma_2 x^{\rho_2}}} < +\infty.$ |
| **Addition** | $(\nu_1, \sigma_1, \rho_1)$ $\oplus$ $(\nu_2, \sigma_2, \rho_2)$ | $\equiv \begin{cases} \max\{(\nu_1, \sigma_1, \rho_1), (\nu_2, \sigma_2, \rho_2)\} & \text{if } \rho_1 \ne \rho_2 \text{ or } \rho_1, \rho_2 < 1 \\ (\nu_1 + \nu_2 + 1, \min\{\sigma_1, \sigma_2\}, 1) & \text{if } \rho_1 = \rho_2 = 1 \\ (\nu_1 + \nu_2 + \frac{2-\rho}{2}, (\sigma_1^{-\frac{1}{\rho-1}} + \sigma_2^{-\frac{1}{\rho-1}})^{1-\rho}, \rho) & \text{if } \rho = \rho_1 = \rho_2 > 1. \end{cases}$ |
| **Powers** | $(\nu, \sigma, \rho)^{\beta}$ | $\equiv (\frac{\nu+1}{\beta} - 1, \sigma, \frac{\rho}{\beta})$ for $\beta > 0$ |
| **Reciprocal\*†** | $(\nu, \sigma, \rho)^{-1}$ | $\equiv \begin{cases} (-\nu - 2, \sigma, -\rho) & \text{if } (\nu+1)/\rho > 0 \text{ and } \rho \ne 0 \\ \mathcal{R}_2 & \text{otherwise} \end{cases}$ |
| **Scalar Multiplication** | $c(\nu, \sigma, \rho)$ | $\equiv (\nu, \sigma|c|^{-\rho}, \rho)$ |
| **Multiplication\*** | $(\nu_1, \sigma_1, \rho_1) \otimes$ $(\nu_2, \sigma_2, \rho_2)$ | $\equiv \begin{cases} \left(\frac{1}{\mu}\left(\frac{\nu_1}{|\rho_1|} + \frac{\nu_2}{|\rho_2|} + \frac{1}{2}\right), \sigma, -\frac{1}{\mu}\right) & \text{if } \rho_1, \rho_2 < 0 \\ \left(\frac{1}{\mu}\left(\frac{\nu_1}{\rho_1} + \frac{\nu_2}{\rho_2} - \frac{1}{2}\right), \sigma, \frac{1}{\mu}\right) & \text{if } \rho_1, \rho_2 > 0 \\ \mathcal{R}_{|\nu_1|} & \text{if } \rho_1 \le 0, \rho_2 > 0 \\ \mathcal{R}_{\min\{|\nu_1|, |\nu_2|\}} & \text{if } \rho_1 = 0, \rho_2 = 0 \end{cases}$ where $\mu = \frac{1}{|\rho_1|} + \frac{1}{|\rho_2|} = \frac{|\rho_1| + |\rho_2|}{|\rho_1 \rho_2|}$ $\sigma = \mu(\sigma_1|\rho_1|)^{\frac{1}{\mu|\rho_1|}} (\sigma_2|\rho_2|)^{\frac{1}{\mu|\rho_2|}}.$ |
| **Product of Densities\*** | $(\nu_1, \sigma_1, \rho_1) \,\&$ $(\nu_2, \sigma_2, \rho_2)$ | $\equiv \begin{cases} (\nu_1 + \nu_2, \sigma_1, \rho_1) & \text{if } \rho_1 < \rho_2 \\ (\nu_1 + \nu_2, \sigma_1 + \sigma_2, \rho) & \text{if } \rho = \rho_1 = \rho_2 \\ (\nu_1 + \nu_2, \sigma_2, \rho_2) & \text{otherwise.} \end{cases}$ |
| **Exponentials\*†** | $\exp(\nu, \sigma, \rho)$ | $\equiv \begin{cases} \mathcal{R}_{\sigma+1} & \text{if } \rho \ge 1 \\ \mathcal{R}_1 & \text{otherwise.} \end{cases}$ |
| **Logarithms\*†** | $\log(\nu, \sigma, \rho)$ | $\equiv \begin{cases} (0, |\nu| - 1, 1) & \text{if } \nu < -1 \\ \mathcal{L} & \text{otherwise.} \end{cases}$ |
| **Functions ($L$-Lipschitz)** | $f(X_1, \dots, X_n) \equiv L \max\{X_1, \dots, X_n\}$ | |

Table 1: ***The Generalized Gamma Algebra.*** Operations on random variables (e.g., $X_1 + X_2$) are viewed as actions on density functions (e.g., convolution $(\nu_1, \sigma_1, \rho_1) \oplus (\nu_2, \sigma_2, \rho_2)$) and the tail parameters of the result are analyzed and reported. In this table, * denotes novel results, and † denotes that additional assumptions are required.

simplicity, we assume that each $f_i = f_{i,k} \circ f_{i,k-1} \circ \cdots \circ f_{i,1}$ where each $f_{ij}$ is an elementary operation in Table 1. To estimate the tail behaviour of $\theta$ conditioned on $X$, we propose an elementary approach involving inverses. For each operation $f_{ij}$, if $f_{ij}$ is a power, reciprocal, or multiplication operation, let $R_{ij}$ be given according to the following:

$$\begin{aligned} \textbf{Powers:} \quad & f_{ij}(x) = x^{\beta}, & R_{ij} &\equiv (1 - \beta, 1, 0) \\ \textbf{Reciprocals:} \quad & f_{ij}(x) = x^{-1}, & R_{ij} &\equiv (2, 1, 0) \qquad \text{and otherwise, let } R_{ij} \equiv 1. \\ \textbf{Multiplication:} \quad & f_{ij}(x, y) = xy, & R_{ij} &\equiv (1, 1, 0) \end{aligned}$$

Letting $f_i^{-1}(x, z)$ denote the inverse of $f_i$ in the first argument, we show in Appendix A,

$$\theta | \boldsymbol{X} = \boldsymbol{x} \equiv \left( \underset{i=1}{\overset{n}{\&}} \, f_i^{-1}(\boldsymbol{x}, Z_i) \right) \& \left( \underset{i,j=1}{\overset{n,k}{\&}} \, R_{ij} \right) \& \, \pi,$$

where $\pi$ denotes the prior for $\theta$. Since the inverse of a composition of operations is a composition of inverses, the tail of $f_i^{-1}(\boldsymbol{x}, Z_i)$ can be determined by backpropagating through the computation graph for $X_i$ and sequentially applying inverse operations. Consequently, the tail behaviour of the posterior distribution for one parameter can be obtained using a single backward pass. Posterior distributions for multiple parameters involve repeating this procedure one parameter at a time, with other parameters fixed.

## 3 Implementation

### 3.1 Compile-time static analysis

To illustrate an implementation of GGA for static analysis, we sketch the operation of the PPL compiler at a high-level and defer to the code in Supplementary Materials for details. A probabilistic program is first inspected using Python's built-in `ast` module and transformed to static single assignment (SSA) form [43]. Next, standard compiler optimizations (e.g., dead code elimination, constant propagation) are applied and an execution of the optimized program is traced [4, 58] and accumulated in a directed acyclic graph representation. A breadth-first type checking pass, as seen in Algorithm 1, completes in linear time, and GGA results may be applied to implement `computeGGA()` using the following steps:

---

**Algorithm 1:** GGA tails static analysis pass

---
**Data:** Abstract syntax tree for a PPL program
**Result:** GGA parameter estimates for all random variables
frontier $\leftarrow$ [rv : Parents(rv) $= \emptyset$];
tails $\leftarrow$ {};
**while** *frontier* $\neq \emptyset$ **do**
    next $\leftarrow$ frontier.popLeft();
    tails[next] $\leftarrow$ computeGGA(next.op,
      next.parent);
    frontier $\leftarrow$ frontier + next.children();
**end**
**return** *tails*

---

- If a node has no parents, then it is an atomic distribution and its tail parameters are known (Table 5);

- Otherwise, the node is an operation taking its potentially stochastic inputs (parents) to its output. Consult Table 1 for the output GGA tails.

### 3.2 Representative distributions

For each $(\nu, \sigma, \rho)$, we make a carefully defined choice of $p$ on $\mathbb{R}$ such that if $X \sim p$, then $X \equiv (\nu, \sigma, \rho)$. This way, any random variable $f(X)$, where $f$ is 1-Lipschitz, will exhibit the correct tail, and so approximations of this form may be used for VI or density estimation. Let $X \equiv (\nu, \sigma, \rho)$ and $0 < \epsilon \ll 1$ denote a small parameter such that tails $e^{-x^\epsilon}$ are deemed to be "very heavy" (we chose $\epsilon = 0.1$). Inspired by ideas from implicit renewal theory [5], our candidate distributions are as follows.

($\rho \leq 0$) If $\rho \leq 0$, then $p_X(x) \sim cx^{-|\nu|}$. One such density is the *Student t distribution*, in this case, with $|\nu| - 1$ degrees of freedom if $\nu < -1$ (generate $X \sim \text{StudentT}(|\nu| - 1)$).

($\rho > \epsilon$) For moderately sized $\rho > 0$, the symmetrization of the generalized Gamma density (2).

($\rho \leq \epsilon$) If $X \equiv (\nu, \sigma, \rho)$ where $\rho$ is small, then $X$ will exhibit much heavier tails, and the generalized Gamma distribution in Case 1 will become challenging to sample from. In these cases, we expect that the tail of $X$ should be well represented by a power law. The generalized Gamma density (Equation (2)) satisfies $\mathbb{E}X^r = \sigma^{-r/\rho}\Gamma(\frac{\nu+1+r}{\rho})/\Gamma(\frac{\nu+1}{\rho})$ for $r > 0$. Let $\alpha > 0$ be such that $\mathbb{E}X^\alpha = 2$. By Markov's inequality, the tail of $X$ satisfies $\mathbb{P}(X > x) \leq 2x^{-\alpha}$. Therefore, we can represent tails of this form by the Student $t$ distribution with $\alpha + 1$ degrees of freedom (generate $X \sim \text{StudentT}(\alpha)$).

### 3.3 Bulk correction by Lipschitz mapping

While a representative distribution will exhibit the desired tails, the target distribution's bulk may be very different from a generalized Gamma and result in poor distributional approximation. To address this, we propose splicing together the tails from a generalized Gamma with a flexible density approximation for the bulk. While many combinations are possible, in this work we rely on the Lipschitz operation in the GGA (Theorem 2) and post-compose neural spline flows [15] (which are identity functions outside of a bounded interval hence 1-Lipschitz) after properly initialized generalized Gamma distributions. Optimizing the parameters of the flow results in good bulk approximation while simultaneously preserving the tail correctness guarantees attained by the GGA.

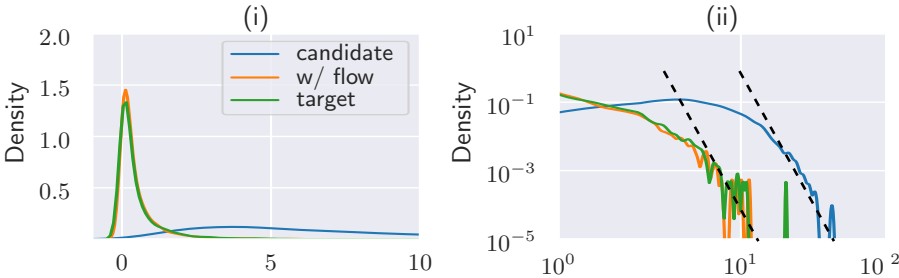

Figure 2: The **candidate** distribution chosen by the GGA calibrates tails to the **target**, but with incorrect bulk. A Lipschitz normalizing **flow** corrects the bulk (i) without changing the tail behaviour, as seen by the parallel tail asymptotics (black dashed lines) in (ii).

**Example 1.** Let $A \in \mathbb{R}^{k \times k}$, $x, y \in \mathbb{R}^k$, with $x_i, y_i, A_{ij} \overset{\text{iid}}{\sim} \mathcal{N}(-1, 1)$. The distribution of $x^\top A y = \sum_{i,j} x_i A_{ij} y_j$ is a convolution of normal-powers [21] and lacks a closed form expression. Using the GGA (Table 1), one can compute its tail parameters to be $(\frac{k}{2} - 1, \frac{3}{2}, \frac{2}{3})$. The candidate given by the GGA representative distribution (Section 3.2) is a gamma distribution with correct tail behaviour, but is a poor approximation otherwise. A learnable Lipschitz bijection is optimized to correct the bulk approximation (Figure 2(i)). From the Lipschitz property, the slope of the tail asymptotics in log-log scale remains the same before and after applying the flow correction (Figure 2(ii)): the tails are *guaranteed* to remain calibrated.

**Example 2.** Consider $\sum_{i=1}^{4} X_i^2$ where $X_i \sim \text{StudentT}(i)$. While we are not aware of a closed-form expression for the density, this example is within the scope of our GGA. Empirical results illustrate that our method (Figure 3(i)) accurately models *both* the bulk and the tail whereas Gaussian-based Lipschitz flows (Figure 3(ii)) inappropriately impose tails which decay too rapidly.

## 4 Experiments

We now demonstrate that GGA-based density estimation yields improvements in tail estimation across several metrics. Our experiments consider normalizing flows initialized from (i) the parametric family defined in Section 3.2 against (ii) a normal distribution (status quo). To further contrast the individual effect of using a GGA base distribution over standard normals against more expressive pushforward maps [15], we also report ablation results where normalizing flows are replaced by affine transforms, as originally proposed in [30]. All experiments are repeated for 100 trials, trained to convergence using the Adam optimizer with manually tuned learning rate. Additional details are available in Appendix D. All target distributions in this section are expressed as generative PPL programs: Cauchy using a reciprocal normal; Chi2 (chi-squared) using a sum of squared normals; IG (Inverse Gamma) using a reciprocal exponential; normal using a sum of normals; and StudentT using a normal and Cauchy ratio. Doing so tasks the static analyzer to infer the target's tails and makes the analysis non-trivial.

Our results in the following tables share a consistent narrative where a GGA base distribution rarely hurts and can significantly help with heavy tailed targets. Standard evaluation metrics such as negative cross-entropy, ELBO, or importance-weighted autoencoder bounds [6] do not evaluate the quality of tail approximations. Instead, we consider diagnostics which do: namely, an estimated tail exponent

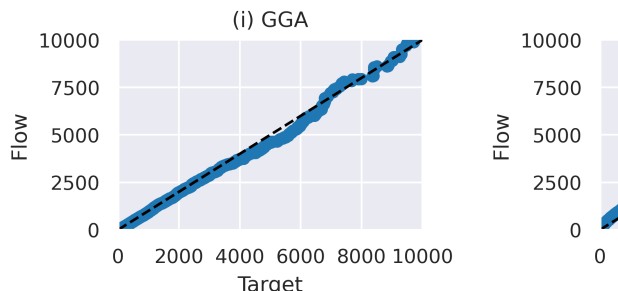

Figure 3: Q-Q plots of density approximations of a heavy-tailed target ($\sum_{i=1}^{4} X_i^2$ where $X_i \sim$ StudentT($i$)) initialized by our GGA candidate (i) and the Gaussian distribution (ii). While the expressive modeling capability of flows enables good approximation of the distribution bulk, Lipschitz transformations of Gaussians inevitably impose miscalibrated squared exponential tails which are not sufficiently heavy as evidenced in (ii).

$\hat{\alpha}$, and the Pareto $\hat{k}$ diagnostic [60]. Except for when targets are truly light tailed ($\alpha = \infty$ in Chi2 and normal), GGA-based approximations are the only ones to reproduce appropriate GPD tail index $\hat{\alpha}$ in density estimation and achieve a passing $\hat{k}$ below $0.2$ in VI. Less surprising is the result that adding a flow improved approximation metrics, as we expect the additional representation flexibility to be beneficial.

**Density Estimation.** Given samples $\{x_i\}_{i=1}^{N}$ from a target density $p$, we minimize a Monte-Carlo estimate of the cross entropy $H(p, q) = -E_p[\log q(X)] \approx -\frac{1}{N} \sum_{i=1}^{N} \log q(x_i)$. The results are shown in Table 2 and Table 3 along with power-law tail index estimates $\hat{\alpha}$ [11]. Closeness between the target Pareto tail index $\alpha$ [11] and its estimate $\hat{\alpha}$ in $q(x)$ suggest calibrated tails. Overall, we see that normal (resp., Cauchy) based flows fails to capture heavy (resp., light) tails, while GGA-based flows yield good tail approximations (lower NLL, $\hat{\alpha}$ closer to target) across all cases.

Table 2: Mean and standard errors (100 trials) of tail parameters $\hat{\alpha}$ (smaller for heavier tails) for various density estimators and targets.

| Target | $\alpha$ | Cauchy ($\alpha = 2$) Flow | GGA Flow | Normal ($\alpha = \infty$) Flow |
|---|---|---|---|---|
| Cauchy | 2 | **2.1 (0.03)** | **2.1 (0.07)** | 7.7 (2.5) |
| IG | 2 | **1.9 (0.03)** | **1.9 (0.092)** | 7.3 (1.7) |
| StudentT | 3 | 2.0 (0.06) | **3.3 (0.45)** | 7.7 (2.3) |
| Chi2 | $\infty$ | 2.1 (0.07) | **5.2 (1.6)** | **6.8 (2.4)** |
| Normal | $\infty$ | 2.9 (0.6) | **8.2 (4.0)** | **8.4 (3.5)** |

Table 3: Mean and standard errors of log-likelihoods $E_p \log q(X)$ for various density estimators and targets. While larger values imply a better overall approximation (row max bolded), log-likelihood is dominated by bulk approximation so these results show that our method (GGA Flow) does not sacrifice bulk approximation quality.

| Target | $\alpha$ | Cauchy ($\alpha = 2$) Flow | GGA Flow | Normal ($\alpha = \infty$) Flow |
|---|---|---|---|---|
| Cauchy | 2 | **−2.53 (0.05)** | −3.22 (0.06) | $-1.2 \times 10^3$ ($6 \times 10^3$) |
| IG | 2 | −3.55 (0.08) | **−3.26 (0.05)** | $-2.6 \times 10^4$ ($6 \times 10^3$) |
| StudentT | 3 | **−2.12 (0.03)** | −2.75 (0.04) | −2.92 (0.47) |
| Chi2 | $\infty$ | −2.30 (0.05) | **−2.03 (0.04)** | −2.24 (0.04) |
| Normal | $\infty$ | −1.53 (0.03) | **−1.41 (0.02)** | −1.42 (0.02) |

**Variational Inference.** For VI, the bulk is corrected through the ELBO optimization objective $E_q \log \frac{p(X)}{q(X)} \approx \frac{1}{N} \sum_{i=1}^{N} \log \frac{p(x_i)}{q(x_i)}$, $x_i \sim q$. Since the density $p$ must also be evaluated, for simplicity, experiments in Table 4 use closed-form marginalized densities for targets. The overall trends also show that GGA yields consistent improvements; the $\hat{k}$ diagnostic [60] indicates VI succeeds ($\hat{k} \leq 0.2$) when a GGA with appropriately matched tails is used and fails ($\hat{k} > 1$) when Gaussian tails are erroneously imposed.

Table 4: Pareto $\hat{k}$ diagnostic ([60]) to assess goodness of fit for VI (mean across 100 trials, standard deviation in parenthesis) on targets of varying tail index (smaller $\alpha$ = heavier tails). A value $> 0.2$ is interpreted as potentially problematic so only values not exceeding it are bolded.

| Target | $\alpha$ | Normal Affine | Normal Flow | GGA Affine | GGA Flow |
|---|---|---|---|---|---|
| Cauchy | $\alpha = 2$ | 0.62 (0.26) | 0.22 (0.059) | 0.68 (0.038) | **0.091 (0.04)** |
| IG | $\alpha = 2$ | 8.6 (1.8) | 8.2 (2.3) | 2.0 (0.4) | 2.9 (0.71) |
| StudentT | $\alpha = 3$ | 1.2 (0.16) | 1.0 (0.43) | 1.5 (0.082) | 1.3 (0.097) |
| Chi2 | $\alpha = \infty$ | 0.57 (0.081) | 0.61 (0.067) | **0.0093 (0.0067)** | **0.089 (0.044)** |
| Normal | $\alpha = \infty$ | 0.53 (0.17) | 0.21 (0.067) | 0.4 (0.086) | **0.2 (0.089)** |

**Bayesian linear regression.** As a practical example of VI applied to posterior distributions, we consider the setting of one-dimensional Bayesian linear regression (BLR) with conjugate priors, defined by the likelihood $y|X, \beta, \sigma \sim \mathcal{N}(X\beta, \sigma^2)$ with a Gaussian prior $\beta|\sigma^2 \sim \mathcal{N}(0, \sigma^2)$ on the coefficients, and an inverse-Gamma prior with parameters $a_0$ and $b_0$ on the residual variance $\sigma^2$. The posterior distribution for $\beta$ conditioned on $\sigma^2$ and $X, y$ is Gaussian. However, conditional on the pair $(X, y)$, $\sigma^2$ is inverse-Gamma distributed with parameters $a_0 + \frac{n}{2}$ and $b_0 + \frac{1}{2}(y^\top y - \mu^\top \Sigma \mu)$, where $\mu = \Sigma^{-1} X^\top X \hat{\beta}$ for $\hat{\beta}$ the least-squares estimator, and $\Sigma = X^\top X + I$. Since $\sigma^2$ is positive, it is typical for PPL implementations to apply an exponential transformation. Hence, a Lipschitz normalising flow starting from a Gaussian initialization will inappropriately approximate the inverse Gamma distributed $p(\sigma^2|X, y)$ with log-normal tails. On the other hand, Lipschitz flows starting from a GGA reference distribution will exhibit the correct tails. We assess this discrepancy in Figure 4 under an affine transformation on four subsampled datasets: `super` (superconductor critical temperature prediction dataset [23] with $n = 256$ and $d = 154$); `who` (life expectancy data from the World Health Organisation in the year 2013 [41] with $n = 130$, $d = 18$); `air` (air quality data [14] with $n = 6941$, $d = 11$); and `blog` (blog feedback prediction dataset [7] with $n = 1024$, $d = 280$). In Figure 4(i), the GGA-based method seems to perfectly fit to the targets, while in Figure 4(ii), the standard Gaussian approach fails to capture the tail behaviour.

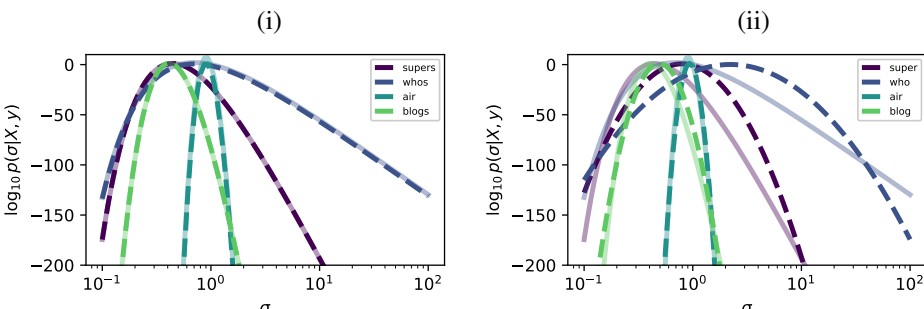

Figure 4: Estimated densities for the posterior distribution of $\sigma^2$ in Bayesian linear regression under optimised exponential + affine transformations from (i) GGA reference, and (ii) Gaussian reference.

**Invariant distribution of SGD.** For inputs $X$ and labels $Y$ from a dataset $\mathcal{D}$, the least squares estimator for linear regression satisfies $\hat{\beta} = \min_\beta \frac{1}{2}\mathbb{E}_{X,Y\sim\mathcal{D}}(Y - X\beta)^2$. To solve for this estimator, one can apply stochastic gradient descent (SGD) sampling over independent $X_k, Y_k \sim \mathcal{D}$ to obtain the sequence of iterations

$$\beta_{k+1} = (I - \delta X_k X_k^\top)\beta_k + \delta Y_k X_k$$

for a step size $\delta > 0$. For large $\delta$, the iterates $\beta_k$ typically exhibit heavy-tailed fluctuations [24]. In this regard, this sequence of iterates has been used as a simple model for more general stochastic optimization dynamics [22, 24]. In particular, generalization performance has been tied to the heaviness of the tails in the iterates [47]. Here, we use our algebra to predict the tail behaviour in a simple one-dimensional setting where $X_k \sim \mathcal{N}(0, \sigma^2)$ and $Y_k \sim \mathcal{N}(0, 1)$. From classical theory [5], it is known that $X_k$ converges in distribution to a power law with tail exponent $\alpha > 0$ satisfying $\mathbb{E}|1 - \delta X_k^2|^\alpha = 1$. In Figure 5, we plot the density of the representative for $\beta_{10^4}$ obtained using our

algebra against a kernel density estimate using $10^6$ samples when $\sigma \in \{0.4, 0.5, 0.6\}$ and $\delta = 2$. In all cases, the density obtained from the algebra provides a surprisingly close fit.

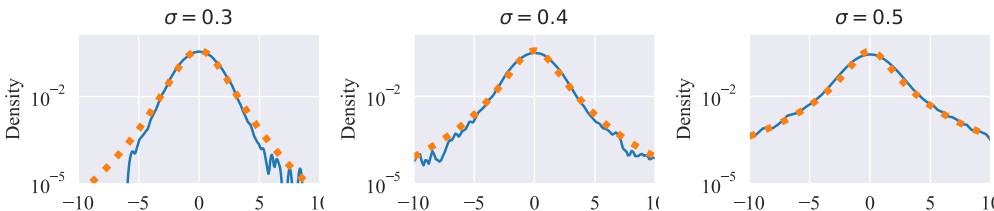

Figure 5: Kernel density estimate of iterates of SGD (blue) vs. GGA predicted tail behaviour (orange)

## 5   Related Work

**Heavy tails and probabilistic machine learning.** For studying heavy tails, methods based on subexponential distributions [18] and generalized Pareto distributions (GPD) (or equivalently, regularly varying distributions [49]) have received significant attention historically. For example, [35] presents closure theorems for regularly varying distributions which are special cases of Proposition 1 and Theorem 2. Heavy tails often have a profound impact on probabilistic machine learning methods: in particular, the observation that density ratios $\frac{p(x)}{q(x)}$ tend to be heavy tailed has resulted in new methods for smoothing importance sampling [53], adaptively modifying divergences [55], and diagnosing VI through the Pareto $\hat{k}$ diagnostic [60]. These works are complementary to our paper, and our reported results include $\hat{k}$ diagnostics for VI and $\hat{\alpha}$ tail index estimates based on GPD.

Our work considers heavy-tailed targets $p(x)$ which is the same setting as [25, 33]. Whereas those respective works lump the tail parameter in as another variational parameter and may be more generally applicable, the GGA may be applied before samples are drawn and leads to perfectly calibrated tails when applicable.

**Probabilistic programming.** PPLs can be broadly characterized by the inference algorithms they support, such as: Gibbs sampling over Bayes nets [13, 48], stochastic control flow [19, 58], deep stochastic VI [4, 52], or Hamiltonian Monte-Carlo [8, 59]. Our implementation target `beanmachine` [50] is a declarative PPL selected due to availability of a PPL compiler and support for static analysis plugins. Similar to [4, 46], it uses PyTorch [40] for GPU tensors and automatic differentiation. Synthesizing an approximating distribution during PPL compilation (Section 3) is also performed in the Stan language by [30] and normalizing flow extensions in [57]. We compare directly against these related density approximators in Section 4.

**Static analysis.** There is a long history of formal methods and probabilistic programming in the literature [26, 29], with much of the research [10] concerned with defining formal semantics and establishing invariants [54] (see [3] for a recent review). Static analysis uses the abstract syntax tree (AST) representation of a program in order to compute invariants (e.g., the return type of a function, the number of classes implementing a trait) without executing the underlying program. It has traditionally been applied in the context of formalizing semantics [29], and has been used to verify probabilistic programs by ensuring termination, bounding random values values [44]. As dynamic analysis in a PPL is less reliable due to non-determinism, static analysis techniques for PPLs become essential. As recent examples, [32] proposes a static analyzer for the Pyro PPL [4] to verify distribution supports and avoid $-\texttt{Inf}$ log probabilities. More relevant to our work are applications of static analysis to improve inference. [38] and [12] both employ static analysis to inform choice of inference method. However, both works do not account for heavy tails whereas the primary goal of GGA-based analysis is to ensure tails are properly modelled.

## 6   Conclusion

In this work, we have proposed a novel systematic approach for conducting tail inferential static PPL analysis. We have done this by defining a heavy-tailed algebra, and by implementing a three-parameter generalized Gamma algebra into a PPL compiler. Initial results are promising, showing that improved inference with simpler approximation families is possible when combined with tail

metadata. While already useful, the generalized Gamma algebra and its implementation currently has some notable limitations:

- The most significant omission to the algebra is classification of log-normal tails. Addition may be treated using [20], but multiplication with log-normal tails remains elusive.

- Since the algebra assumes independence, handling of dependencies between defined random variables must be conducted externally. This can be addressed using a symbolic package to decompose complex expressions into operations on independent random variables.

- Scale coefficients $\sigma$ for conditional distributions may often be inexact, as exact marginalization in general is NP-hard [28]. Treatment of disintegration using symbolic manipulations is a significant open problem, with some basic developments [9, 45].

- Compile-time static analysis is only applicable to fixed model structures. Control flow, open-universe models [36], and PPLs to support them [4] are an important future research direction.

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
