# OpenReview forum: "A Heavy-Tailed Algebra for Probabilistic Programming"
_NeurIPS.cc/2023/Conference — NeurIPS 2023 poster_

### Official Review · Reviewer_bd7W · 2023-07-04

**Soundness:** 4 excellent
**Presentation:** 4 excellent
**Contribution:** 3 good
**Rating:** 7
**Confidence:** 3

**Summary:**

The paper proposes a static analysis technique for probabilistic programming languages, which annotates random variables with metadata characterizing their tail behavior. In particular, generalized Gamma distributions are used for this purpose. It is shown that they are closed under a number of operations, including addition, multiplication, and, under some conditions, reciprocals. This gives rise to an algebra which is used to statically infer the tail behavior of random variables throughout the probabilistic program, including posterior distributions. At runtime, the random variables are then estimated by neural splice flows initialized to the inferred Gamma distributions, which guarantees the correct tail behavior while allowing flexibility for the bulk of the probability mass. It is shown that this approach yields the correct behavior in many instances in which conventional methods fail.

**Strengths:**

 - The paper addresses an important and interesting problem in probabilistic programming.
 - The static analysis approach to analyzing tail behavior is novel to my knowledge.
 - The chosen approach is competently executed: The provided algebra covers a wide variety of distribution types and operations, many of which utilizing newly derived theoretical results.
 - Combining the inferred Gamma distribution parameters with neural splice flows is a clever way to maintain the correct tail behavior while allowing flexible estimation of the distribution's bulk.
 - Encouraging experimental results are provided for a number of tasks, including density estimation, variational inference, and Bayesian linear regression.
 - The paper is well written and easy to follow. Detailed derivations for the new theoretical results are provided in the appendix.


**Weaknesses:**

- The models used for the experimental results are somewhat small. While simple monovariate distributions are fine to get a sense for the behavior of the method, it would be good to include some larger programs to demonstrate the robustness of the method.
- The paper mentions a number of limitations of the approach: It does not cover log-normal tails, operations between dependent variables, or dynamic model structures. Given these limitations, it would be especially useful to highlight some set of more complex models used in practice to which the method is nevertheless applicable.

**Questions:**

- How does the system behave when the assumptions discussed above are not fully met, e.g., when there is an operation between dependent variables? Is it possible to make use of partial results for a given program?
- The discussion on posterior distributions mostly focuses on two random variable connected by a chain of operations, but notes that other cases may also be covered. Is it really the case that any posterior distribution may be estimated this straightforwardly? Can the system as implemented cover complicated multivariate computation graphs?

**Limitations:**

The limitations have been well addressed, though their practical implications could be stated more directly (as discussed above).

---

> ### Author Rebuttal · Authors · 2023-08-07
>
> Thank you for the positive assessment of our work!
>
> > How does the system behave when the assumptions discussed above are not fully met, e.g., when there is an operation between dependent variables? Is it possible to make use of partial results for a given program?
>
> > The discussion on posterior distributions mostly focuses on two random variable connected by a chain of operations, but notes that other cases may also be covered. Is it really the case that any posterior distribution may be estimated this straightforwardly? Can the system as implemented cover complicated multivariate computation graphs?
>
> This relates to questions raised by Reviewer 9rGq's --- in a nutshell, handling dependencies accurately will require symbolic treatment.
> You are right that not all posteriors are covered, and it is actually NP-hard (line 274) to cover all possible posteriors. While the GGA is capable of analyzing some complicated multivaraite computation graphs, it is currently limited to computation graphs satisfying (or as the discussion around __independence assumptions__ shows,
> can be equivalently rewritten to satisfy) the independence assumption for the input operands of every GGA operation.
> Generally speaking, to exactly determine tail behavior at scale is likely to require further advancements on top of what we propose.
> However, we do agree that there is still partial utility when results are violated and will include this discussion in the revision.
> In particular, since the GGA is quite simple, there is nothing preventing its usage at scale to provide an estimate of the tail behavior. To our knowledge, there is no alternative in the literature, and the GGA may reveal information about the target density (e.g. to diagnose issues) that would otherwise be opaque.
>
> > ...the method's evaluation is limited to unimodal distributions. The density estimation experiments have been done on standard distributions. I am unsure about whether this is a limitation of the evaluation or a limitation of the proposed method.
>
> We believe evaluation against a multi-modal mixture target is reasonable provided the mixture is given explicitly (e.g. as a list of component random variables and their mixture probabilities) so that a PPL compiler can analyze it statically.
> Due to space constraints, we did not include an operation for a mixture of densities in the GGA although such a result is possible within the GGA. Indeed, denoting a mixture of densities by $\cup$, this operation would be added to Table 1 in the form
> $$
> (\nu_1,\sigma_1,\rho_1) \cup (\nu_2,\sigma_2,\rho_2) \equiv \max\{(\nu_1,\sigma_1,\rho_1),(\nu_2,\sigma_2,\rho_2)\}.
> $$
> At this point, the framework is capable of naturally handling multi-modality through its
> normalizing flow bulk adjustment.
> For example, (Wehenkel \& Louppe, 2019) or (Durkan et al., 2019) both provide multi-modal density
> approximations and could be used as the normalizing flow.
> Since multimodality is a bulk property (e.g. a Gaussian mixture with finitely many components continues to have Gaussian tails), it does not affect tail asymptotics
> and the GGA remains applicable.
> With this in mind, multimodality is not really an assessment of the GGA, and so we focussed our attention on unimodal distributions with clearer comparisons. We are happy to include a multimodal example in the revised version, however.

---

> > ### Comment · Reviewer_bd7W · 2023-08-15
> > **Response to Rebuttal**
> >
> > Thank you for your detailed response. I appreciate the clarifications, and am happy to keep my score.

---

### Official Review · Reviewer_mtnU · 2023-07-05

**Soundness:** 3 good
**Presentation:** 3 good
**Contribution:** 3 good
**Rating:** 7
**Confidence:** 3

**Summary:**

The paper develops an algebra which acts on a three-parameter family of tail asymptotics based on the generalized Gamma distribution. The algebraic operations are closed under addition, multiplication, powers, and a full list is given in Table 1. With this algebra, tail calculation can be done automatically in probabilistic programming instead of analytically. The paper also proposed a method of splicing bulk and calibrated tails with experimental verification.

**Strengths:**

1. The paper develops a heavy-tailed algebra for probabilistic programming and also proposed a method of splicing bulk and calibrated tails with experimental verification.
2. Example 4 in the appendix had a calculation for the product of two random variables.

**Weaknesses:**

I did not notice any glaring weaknesses in the paper but I am also not a subject matter expert on probabilistic programming.

**Questions:**

1. In the case of multi-dimensional random variables and dependent tails, would you consider using an extreme-value copula?
2. In the case of dependent tails and bulk to tail splicing/ calibration/ extrapolation, would you consider using an Archimax copula?

**Limitations:**

The limitations were clearly defined and addressed in the paper.

---

> ### Author Rebuttal · Authors · 2023-08-07
>
> Thank you for reading and reviewing our paper. Since our work only focuses on univariate tails, we agree that copulas provide a promising direction which can be combined with other work applying normalizing flows to multi-variate heavy tails (ATAF paper) in order to improve multivariate heavy-tailed approximation. The choice of copula is indeed important here, so the extension will require further work.

---

> > ### Comment · Reviewer_mtnU · 2023-08-15
> >
> > Thanks for the response. I have read all the other reviews and responses and increased the score to 7.

---

### Official Review · Reviewer_E196 · 2023-07-06

**Soundness:** 4 excellent
**Presentation:** 3 good
**Contribution:** 4 excellent
**Rating:** 7
**Confidence:** 3

**Summary:**

During inference, we are often interested in the behavior of the tails of the distributions we are analyzing. Heavy or light tails may necessitate switching algorithms so that inference remains stable for example. This paper describes a calculus by which a probabilistic programming language may calculate the tails of distribution objects derived from operations performed on elemental distributions. They apply their method to perform some simple probabilistic programming applications.

**Strengths:**

The authors apply their method to a huge number of distributions and their transformations. They summarize their rules helpfully in charts. It seems as though their method may also simply be added to other probabilistic programming languages. I would certainly appreciate being able to call the tail parameters of a distribution object in Pyro for example.

**Weaknesses:**

The authors in section 3 suggest a method for fitting a density that consists of fitting its tail using their calculus and then fitting the bulk using a flow model. This is a reasonable choice for their simple experiments and there are a number of other choices they could have made to fit the bulk. The exposition of the paper sets it up in figure 1 and section 3 as essential to their probabilistic programming method, but I don't believe that to be that case. It would clarify and strengthen the paper if authors clarified the relation of their flow method for estimating densities to the rest of their contributions.

**Questions:**

In line 134 you describe the representative distribution of a distribution with a tail with a small $\rho$. You motivate your construction by matching moments, but could you describe more rigorously why this is a reasonable

I also wonder why the authors don't consider incorporating discrete distributions such as Poisson and Geometric distributions, which can be treated as continuous in their setting, say by adding a uniform random variable, so a geometric would have a $(0, 0, \log(p))$ tail for example. There are often cases where discrete data is treated as continuous, such as for RNAseq counts.

**Limitations:**

The authors describe limitations in section 6.

---

> ### Author Rebuttal · Authors · 2023-08-07
>
> Thank you for the positive assessment of our work! Yes, you are correct that the choice of spliced flow is non-essential. Our intention was to propose one such construction which was sufficiently flexible to capture the bulk while also respecting tail asymptotics computed by the GGA and we achieved this by proving Theorem 2 and operationalizing it in Section 3. Alternative constructions which preserve tail asymptotics may also perform well and we will clarify our contributions section to reflect this in the revision.
>
> > In line 134 you describe the representative distribution of a distribution with a tail with a small . You motivate your construction by matching moments, but could you describe more rigorously why this is a reasonable
>
> It is known that very heavy subexponential distributions (small $\rho$)---which often come from repeated multiplications---closely resemble a power law in practice, as shown in line 522. If we do not approximate this case by a power law, the representative generalized Gamma distribution often provides a very poor approximation to the bulk, inhibiting the final stage of bulk correction. The moment-matching approach is loosely derived from ideas in implicit renewal theory (Buraczewski, Dariusz, Damek, Mikosch 2016, which we are happy to cite in the revised version) and we found it to perform the best in practice.
>
> __Incorporating discretes__: Thank you for the suggestion! Yes, as you have already noted with the geometric distribution (although the class would be $(0,\log p, 1)$), there is nothing to prohibit the use of the GGA for operating on discrete distributions by treating them as continuous through a suitable convolution as suggested.

---

> > ### Comment · Reviewer_E196 · 2023-08-15
> > **Response to authors**
> >
> > I thank the authors for their response, they have answered my questions.

---

### Official Review · Reviewer_9rGq · 2023-07-06

**Soundness:** 3 good
**Presentation:** 3 good
**Contribution:** 4 excellent
**Rating:** 7
**Confidence:** 4

**Summary:**

The paper addresses the problem of density estimation of probabilistic models (expressed as probabilistic programs), with a focus on their tails. This is important for several Bayesian inference methods: importance sampling can exhibit infinite variance if the proposal has a lighter tail than the target, many black box variational inference methods cannot change their tail behavior and MCMC algorithms may lose ergodicity if the tail of the target is outside a particular class. This paper proposes a static analysis pass on probabilistic programs consisting of sampling from primitive distributions and arithmetic operations on the sampled variables (addition, multiplication, division, logarithm, exponential, and Lipschitz functions). The analysis is based on generalized Gamma distributions, which capture many tail behaviors of primitive distributions (e.g. Gaussian, Gamma, Weibull, Student-t, Pareto, but not the log-normal) and are closed under the above operations on random variables (or can be bounded if not represented exactly). As a consequence, a compiler pass can compute the parameters of a generalized Gamma distribution with the same (or at least providing a bound on the) tail behavior as the posterior distribution of the probabilistic program. Since the bulk of this distribution may be very different from the posterior, the paper combines it with neural spline flows: effectively, this computes the pushforward distribution under a Lipschitz function, resulting in a better bulk approximation while preserving the tails.
The experimental evaluation demonstrates that this approach to density estimation usually improves the tail estimation compared to status-quo initializations with a normal or Cauchy distribution, both for density estimation using normalizing flows and variational inference, where the target distributions are simple Cauchy, Inverse Gamma, Student-t, χ^2 and normal distributions. Similarly, the density estimation of the variance parameter of a Bayesian linear regression model is improved compared to the standard Gaussian approach. Finally, they observe that their method (without normalizing flows) provides a very close fit for the density estimation of the iterates of stochastic gradient descent applied to a least-squares linear regression.

**Strengths:**

The paper's main idea of using a static analysis to compute the tails of the posterior distribution of a probabilistic program is very interesting because tails are difficult to estimate. Such an ahead-of-time analysis with correctness guarantees is therefore very useful. In order to arrive at this result, the paper proves several new theoretical results on generalized Gamma distributions, which may be of independent interest. The experimental results show that in combination with normalizing flows, this often performs better than existing methods that assume a Gaussian base distribution. The presentation of the paper is generally clear and readable.

Overall, this paper has an innovative idea with promising results, which are of interest to the probabilistic programming community.

**Weaknesses:**

While the paper's idea is very interesting, I have a few concerns about the soundness and presentation. I will update my rating if the author's answers address my concerns.

Guarantees and assumptions: What exactly are the guarantees (mentioned in line 149) and assumptions of the static analysis? It would be good if this could be stated in one place, ideally as a self-contained theorem. Table 1 mentions that "additional assumptions are required" for several operations, but I was not able to find them in the paper. The assumption that the operands are independent is also very important and only really mentioned at the very end (line 270). It is also unclear what exactly the guarantees are for programs involving reciprocals, exponentials, and logarithms because the generalized Gamma distributions are not closed under these operations. In this case, will the computed tail be an upper or a lower bound? What happens if such operations are applied repeatedly? While there are theorems and proofs for each individual operation, I was not able to find a theorem clarifying the assumptions and guarantees about the composition of such operations, especially if the resulting tail cannot be represented exactly in the GGA.

Bulk correction: Section 3.3 (the bulk correction via normalizing flows) is informal and light on details. I assume the neural spine flows preserve the tails because they are 1-Lipschitz maps? This is not obvious to me, and I think it would be good if the paper could elaborate on why the normalizing flows don't affect the tails. Again, putting the statement in the form of a theorem with all necessary assumptions would be helpful.

Probabilistic programming language: The grammar of the probabilistic programming language is only introduced by example, there is no comprehensive description, not even in the appendix. Furthermore, the name "programming language" sounds like there is support for control structure (branches, loops), which doesn't seem to be the case in this approach. Rather, the "programming language" seems to be a simple expression language with the grammar `E ::= sample(D) | op(E, ..., E)` where `E` is an expression, `op` is a supported operation, and `D` is a supported primitive distribution. It would be good if this could be clarified in the paper.

Independence assumption: a main assumption is that the operands to all operations are independent random variables. This seems to be difficult to guarantee in practice because posteriors of parameters are often correlated in probabilistic models. Does this mean that (for now), all operands of all operations need to be hand-checked for independence? Does this severely restrict the class of probabilistic models that can be handled by this approach? If so, this should be discussed. It is only briefly mentioned in line 270.

The following points are weaknesses, but less important than the previous points:

Experimental evaluation: the experiments demonstrate benefits for very simple benchmarks (one-dimensional primitive distributions, one-dimensional Bayesian linear regression). If more realistic experiments (e.g. higher-dimensional examples) could be performed, this would strengthen the paper. However, I don't consider this a condition for publication because the results are already interesting.

Abstract interpretation: the static analysis pass, as I understand it, performs abstract interpretation (a common technique in static analysis) where the abstract domain is given by the generalized Gamma algebra. It would be good to mention this keyword, as it helps put this method into context.

**Questions:**

- Line 92 claims that closure under some operations is known. Could you provide references for this?
- What exactly are the guarantees and assumptions for operations under which GGA is not closed? What happens if such operations are applied repeatedly? (cf. Weaknesses)
- The operations in Table 1 only apply to equivalence classes $(\nu, \sigma, \rho)$. What happens if the result of an operation is $\mathcal L$ or $\mathcal R_1$? Will those be treated as $(?, ?, \infty)$ or $(-1, ?, 0)$ in subsequent operations? If so, what values are assumed for the question marks? Why don't the infinities lead to problems?
- Section 3.3: could you elaborate on why the normalizing flows don't affect the tails (cf. Weaknesses)?
- In Table 4, GGA Flow does not seem to work for IG and StudentT. Do you know what the reason for this is? It seems unexpected given all the claimed benefits.

Typos:
- Equation (1): should be $= cx^\nu ...$ or $\sim x^\nu ...$
- line 111: missing indices $f_{ij}$, $R_{ij}$.
- line 131: shouldn't it say $\rho \le 0$ instead of $\rho \le -1$?
- line 201: it is unclear where the "conditional on" clause ends, please rephrase


**Limitations:**

Limitations are discussed in the Conclusion. I believe a few items should be added to this list:
- no control structure in the programming language (it is an expression language, cf. Weaknesses)
- tails are not exact (only bounds) if certain operations are involved

---

> ### Author Rebuttal · Authors · 2023-08-07
>
>
> Thank you for taking the time to provide a detailed review and giving us the opportunity to address your concerns. We will discuss each point not addressed in our overall author response sequentially.
>
> ## Weaknesses
>
> **Guarantees and assumptions**:
> Thank you for the excellent suggestion! To improve readability of Appendix A in the revision, we will book-end the results with the following main theorem.
>
> **Main Theorem**: Let $X_1,\dots,X_n$ be independent random variables with generalized Gamma tails.
> Then
> - For any $i \neq j$, $c \neq 0$ and $\beta > 0$, $X_i + X_j$, $X_i \times X_j$, $c X_i$, and $X_i^\beta$ also exhibit generalized Gamma tails as detailed in Table 1 (addition, multiplication, scalar multiplication, powers).
> - If $X_i$ and $X_j$ have densities $p_i$ and $p_j$, then the random variable with density proportional to $p_i p_j$ has generalized Gamma tails as in Table 1 (product of densities).
> - If $X_i$ is generalized Gamma distributed, then $\frac{1}{X_i}$ has _exactly_ the generalized Gamma distribution described in Table 1 (reciprocals).
> - If $X_i$ has density $p$ continuous at zero with $p(0) > 0$, then $\frac{1}{X_i}$ also has generalized Gamma tails described in Table 1 (reciprocals).
> - For any Lipschitz function $f$, the tail of $f(X_1,\dots,X_n)$ is no heavier than the generalized Gamma tail detailed in Table 1 (Lipschitz functions). If $f$ is asymptotically linear then $f(X_1,\dots,X_n)$ has _exactly_ the described tail.
> - $\log X_i$ and $e^{X_i}$ have tails no heavier than the generalized Gamma tails in Table 1 (exponentials, logarithms). If $X_i$ is regularly varying, then $\log X_i$ has  _exactly_ the described tail. If $X_i$ is exponentially distributed, then $\exp X_i$ has _exactly_ the described tail.
>
> Hopefully this clears up any confusion around the theoretical results.
>
> **Repeated application**:
> Provided independence holds, composition of operations in the GGA remain consistent unless one applies Lipschitz functions, logarithms, or exponentials. If one of these operations is applied, the tail becomes an upper bound, which remains consistent under addition, multiplication, and powers, but not reciprocals. Given that we are working with a fixed class of tails, such behavior is inevitable, and it is possible to perform a sequence of operations for which the tail no longer becomes accurate. Nevertheless, we have endeavored to create a system to balance theoretical consistency with practical value, and find that in most cases, the GGA algebra produces a tail that is either exact, or a very good estimate (the SGD example was particularly convincing to us).
>
> **Bulk correction**: Yes, neural spline flows are 1-Lipschitz linear functions in the tail. We implied this by writing "identity functions outside of a bounded interval" (line 147) but will revise to be more explicit for clarity.
>
> **Probabilistic programming languages**: We would like to point out the distinction between the host PPL and the symbolic expressions where the GGA is applicable. The GGA itself is not a PPL, but rather an ahead-of-time analysis method compatible with a PPL. Our host PPL (beanmachine) is based in Python and capable of branching and looping. At the end of a single execution trace, the expressions constituting a target random variable is analyzed by the GGA to determine its tail parameters.
>
> Perhaps more interesting, and indeed outside of the scope of the GGA we have developed, is the ability to directly analyze control flow structures. However, we note that analysis of a mixture random variable generated by branching control flows requires marginalization where even approximate inference is known to be NP-hard (line 274 and Koller and Friedman, 2009). Additional assumptions such as junction-tree structure or restricted distribution families may permit further analysis and we leave this for future work.
>
> **Experimental evaluation**: We have restricted our experiments to those where we know the solution exactly, as it is difficult to provide a good frame of reference otherwise.
>
> **Abstract interpretation**: Thank you for this keyword, indeed, this describes our approach very well. We will include it in the revised version.
>
> ## Questions
>
> Responses to questions not already addressed by the "Weaknesses" section:
>
>  - Line 92 references --- These results are cited in Appendix A, but we are happy to include these citations here as well.
>  - Table 1 $\mathcal{L}_1$ or $\mathcal{R}_1$ results --- Again, we are taking some liberties to encode distributions that do not fit into the generalized Gamma tail framework. $\mathcal{L}$ is treated as $(0,1,\infty)$ and $\mathcal{R}_1$ is treated as $(-1,1,0)$. Any operation which results in a super-light tail ($\rho = \infty$) becomes $\mathcal{L}$ and any operation which results in a super-heavy tail ($\rho \leq 0$ and $\nu \geq -1$) becomes $\mathcal{R}_1$. Each of the operations are consistent under these definitions, where $1/\infty$ is treated as $0$. One deficiency here that we recently became aware of is that multiplication with a constant is not always equal to multiplication by $\mathcal{L}$ --- this is the one operation where multiplication by $\mathcal{L}$ can give different answers depending on how $\nu,\sigma$ is defined. This is unavoidable, but our procedure still offers an excellent heuristic even in this setting. In any case, we will include this discussion in the revised version.
> - Table 4 IG and StudentT --- The Pareto $\hat{k}$ diagnostic (Yao, Vehtari, Simpson, Gelman, 2018) is based on a heuristic argument and as a test for divergence is known to yield false positives, which may be the  case in the two examples mentioned.

---

> > ### Comment · Reviewer_9rGq · 2023-08-15
> >
> > Thank you for the thorough response.
> >
> > Regarding the point about the guarantees & assumptions as a self-contained theorem, I wasn't looking for a theorem describing Table 1, but a theorem about the tails of a whole probabilistic program, not just individual operations. I don't mind the individual operations described in a table (as long as the assumptions are written down!). My concern was mostly about the composition of operations and the guarantees that still hold for such compositions.
> >
> > The paragraph "repeated applications" in your rebuttal is what I was looking for. Such a discussion should definitely be included in the paper. It would be even better if this could be turned into a theorem saying under what circumstances (i.e. which operations are allowed to occur in the program) and assumptions (e.g. independence), the computed tail of a *whole* probabilistic program (not just individual operations) is exact, an upper bound, a lower bound, or just an approximation. For the most part, my concerns regarding the theoretical soundness have been alleviated though.
> >
> > Thank you for your other clarifications; they make sense to me. Overall, I have updated my score from 5 to 7.

---

### Author Rebuttal · Authors · 2023-08-07

We thank the reviewers for their time and valuable feedback. We appreciate that reviewers have recognized the novelty of our approach and its applications in ahead-of-time static analysis of probabilistic programming languages (PPL). Suggested minor changes and fixes have been incorporated into the working document. Below we respond to points shared by more than one reviewer and provide more specific responses to individual reviewers as comments.

**Independence assumption**: Independence of operands is evidently key to the algebra functioning correctly, as for example, the tail of $X^2$ differs from the tail of $X Y$ even if $X$ and $Y$ share the same tail. We will include a few words in Section 2 to be clear about this. To deal with arbitrary dependencies that arise due to composition requires a combination of symbolic methods with the GGA. For example, if $Y = X_1 + X_2$ and $Z = X_2 + X_3$, then even if $X_i$ are independent the GGA cannot be applied as-is to $Y + Z$. However, the equivalent symbolic representation $Y + Z = X_1 + 2 X_2 + X_3$ can be dealt with using the GGA. This is discussed in the conclusion (line 270), but we can include further discussion on this matter in the revision.

---

### Decision · Program_Chairs · 2023-09-21

**Decision:**

Accept (poster)

**Comment:**

The authors give a systematic approach to analyzing the tail of random variables in the static analysis approach of a probabilistic programming language compiler, with accompanying empirical results. This is an important question as the tail behavior influences the choice/performance of MCMC and variational inference procedures. In response to reviewer questions, authors explicit stating guarantees for how tails of random variables transform under a catalogue of operations, which clarifies the theoretical grounding for their work. Limitations include the restriction to univariate distributions and independence assumptions, but the paper remains an excellent initial foray into the problem.

Reviewers gave uniformly high rating after discussion. I recommend acceptance.